# Closed-form proximal operator of regularized exponential functions for incremental learning

## Abstract

Incremental model-based minimization methods have recently been proposed as a way to mitigate numerical challenges associated with stochastic or online optimization. One of the main desirable properties is stability w.r.t. step-size choice and loss-function weights. Such properties make them desirable for use-cases when tuning parameters is prohibitive. In contrast to incremental gradient methods, the main computational tool is the proximal operator, rather than the gradient. And this operator is exactly one of the main gaps for adoption in practice - it may be both inefficient in practice, and harder to implement for a practitioner due to the lack of closed-form formulas and expressive calculus.

In this work, we aim to address this challenge for a specific family of losses, which are a composition of exponential on linear functions. One prominent application in mind is that of Poisson regression, where the negative log-likelihood is of this form. We devise a closed-form formula for the proximal operator in terms of Lambert's W function, whose implementation is available in many standard numerical computing and machine-learning packages, such as SciPy or TensorFlow. Then, we show that expressing the same formula in terms of the less-known Wright-Omega function, that is also available in SciPy, provides substantial numerical benefits. Finally, we provide an open-source vectorized PyTorch implementation of the Wright-Omega function and the proximal operator, ported from SciPy. This allows practitioners wishing to use the algorithm devised here to use the entire arsenal of tools provided by PyTorch, such as automatic differentiation and GPU computing. We have made our code available at `https://anonymous.4open.science/r/exponential-proximal-point-B8DD`.

## 1 Introduction

In modern machine learning, algorithms are used to *incrementally* minimize either the sum or expectation of functions of the form $h(\boldsymbol{w}, \boldsymbol{x})$, where the samples $\boldsymbol{x}$ come from a training set or are sampled from a distribution. Typically, the vector $\boldsymbol{w}$ denotes the parameters of the model we wish to learn, and $h$ denotes the cost of mis-prediction. In both incremental paradigms, stochastic and online, the learning algorithm iteratively updates the current estimate of the parameters $\boldsymbol{w}$ based on the arriving samples $\boldsymbol{x}$.

The most popular *incremental*[1] methods ranging from stochastic (Robbins & Monro, 1951) and online (Zinkevich, 2003; Gordon, 1999) gradient methods, AdaGrad (Duchi et al., 2011; McMahan & Streeter, 2010), AdamW (Loshchilov & Hutter, 2019), and others, use first-order information about the (sub) gradients of the function $h$ w.r.t. $\boldsymbol{w}$. These methods, while attractive either theoretically or in practice, require careful tuning of their step-size, or learning rate. Wrong step-size selection may lead to sub-optimal performance at best, or even divergence and floating-point over-flow.

As an example, consider the problem of regularized Poisson regression (Nelder, 1974) for predicting the conditional mean $\mathbb{E}[y|\boldsymbol{\phi}]$ where $y|\boldsymbol{\phi} \sim \mathrm{Poisson}(\langle \boldsymbol{w}, \boldsymbol{\phi} \rangle)$ by minimizing the regularized negative log-likelihood

---

[1]Since this paper's focus is on the computational aspect, rather than the theoretical convergence properties, we shall refer to both the stochastic and online methods as incremental methods.

over the training samples:

$$\frac{1}{m}\sum_{i=1}^{m}(\exp(\langle \boldsymbol{w}, \boldsymbol{\phi}_i\rangle) - y_i\langle \boldsymbol{w}, \boldsymbol{\phi}_i\rangle) + \frac{\alpha}{2}\|\boldsymbol{w}\|_2^2.$$

We immediately see that the summand gradients involve the exponential function, and therefore may be of a very large magnitude. With a step-size slightly too large, the model parameters will quickly grow in norm and cause a floating point overflow when we attempt to compute gradients for future training samples.

The recently devised model-based minimization methods, proposed by Asi & Duchi (2019); Davis & Drusvyatskiy (2019); Kulis & Bartlett (2010), are significantly more robust w.r.t. the step-size choice. Such algorithms are useful when either step-size tuning is overly expensive, or when the algorithm needs to run for a prolonged period of time without human supervision. These methods assume that $\ell$ is approximated by some model[2] $f$, which may depend on the current parameters $\boldsymbol{w}$ and the data point $\boldsymbol{x}$, and compute the updated parameters $\boldsymbol{w}^+$ using the *proximal operator* (Moreau, 1962; 1965):

$$\boldsymbol{w}^+ = \operatorname{prox}_{\eta f}(\boldsymbol{w}) \equiv \arg\min_{\boldsymbol{u}}\left\{f(\boldsymbol{u}) + \frac{1}{2\eta}\|\boldsymbol{u} - \boldsymbol{w}\|_2^2\right\}.$$

Intuitively, the idea is that our updated parameters balance minimizing the approximation of the cost, and staying close to the current parameters. This balance is determined by the step-size $\eta$.

When $f$ is the first-order Taylor approximation of the cost of mis-prediction $h$, we recover the classical incremental gradient method. Moreover, when the model is the cost function itself, namely, $f \equiv h$, we obtain the incremental proximal-point method. In this case, we use the structure of the entire cost function $h$ rather than just its slope. Finally, in the vast majority of cases, the cost is a composition of a loss $\ell$ onto a machine-learned function $m$, namely, $h \equiv \ell \circ m$. In this case, we can construct a model of $h$ by composing $\ell$ onto a first-order Taylor approximation of $m$. In this manner we obtain the Gauss-Newton or the prox-linear method. Note, that in this formulation $m$ may be as complex as we desire, such as a neural network. These possible models, among others, were all studied in Asi & Duchi (2019); Davis & Drusvyatskiy (2019) and references therein.

In general, computing $\operatorname{prox}_{\eta f}$ heavily depends on the structure of $f$, and there is no efficient closed-form solution. Indeed, $f$ may be as complex as we desire. Thus, the main challenge, both in terms of computational efficiency and usefulness in practice, lies in an easy to implement and efficient proximal operator.

In this work we do not propose yet another such algorithm, but rather deal with the implementation such algorithms for specific functions that appear in machine learning applications. We propose an implementation for model functions of the form:

$$f(\boldsymbol{w}; \boldsymbol{\theta}, \boldsymbol{\phi}, b, \alpha) = \exp(\langle \boldsymbol{\theta}, \boldsymbol{w}\rangle + b) + \langle \boldsymbol{\phi}, \boldsymbol{w}\rangle + \frac{\alpha}{2}\|\boldsymbol{w}\|_2^2. \tag{1}$$

By the notation $f(\boldsymbol{w}; \boldsymbol{\theta}, \boldsymbol{\phi}, b, \alpha)$ we mean a function of $\boldsymbol{w}$ parametrized by the remaining arguments. We use this notation since we study the minimization w.r.t. $\boldsymbol{w}$, and therefore we treat the remaining arguments as parameters. Our main application in mind is Poisson regression, where the regularized negative log-likelihood function w.r.t. each sample is a special case of this form. The underlying Poisson regression model does not have to be linear, and in this case the prox-linear approach discussed in Davis & Drusvyatskiy (2019); Asi & Duchi (2019) yields functions of this form via the first-order approximation of the underlying model. Nonetheless, additional applications exist, such as minimizing the exponentially tilted loss (Li et al., 2023), by transforming the minimization problem to an equivalent one using exponential functions, as described in Shtoff (2024b).

The exponential function in the cost induces a difficulty for gradient methods. From a theoretical perspective, the cost and all its derivatives are unbounded, even if the parameters $\boldsymbol{\theta}, \boldsymbol{\phi}, b, a$ are bounded. From a practical perspective, we may encounter floating-point overflows during training due to the exponentiation of possible large numbers. As shown in Asi & Duchi (2019), the model based minimization framework does not suffer

---

[2]in this context, a "model" is not a machine-learned model, but rather a family of approximating functions that attempt to model the cost $\ell$

from these theoretical issues if the training data that is manifested in the parameters $\boldsymbol{\theta}, \boldsymbol{\phi}, b$ is properly normalized. Moreover, as we shall see in this paper, the framework also does not suffer from the numerical issues of over-flows. Thus, our work facilitates a more reliable training procedure when training with cost functions involving exponentiation.

The work of Shtoff (2024a) develops several frameworks facilitating efficient and easy to implement algorithms for computing proximal operators of many cost function families useful in machine learning, and utilizes the framework to devise algorithms and code for many concrete examples. Our paper can be seen as a direct application of one of the frameworks devised in Shtoff (2024a) to tackle the family described in equation 1.

The main contributions of our work are:

- a closed-form formula for $\mathrm{prox}_{\eta f}$, where $f$ is of the form in equation 1, in terms of the well-known Lambert-W function;

- a reformulation of the above formula using the Wright-Omega function (Corless & Jeffrey, 2002), available in SciPy, to obtain a more numerically-favorable formula;

- a vectorized implementation of the Wright-Omega function in PyTorch (Ansel et al., 2024), and a vectorized implementation of $\mathrm{prox}_{\eta f}$ in both PyTorch and SciPy.

## 2   Related work

The proximal operator of the entropy function $x \to x \ln(x) - x$ has a known formula (Combettes & Pesquet, 2011, table 2) in terms of functions of the form $x \to W(a \exp(bx))$. The proximal operator $x \to \exp(x)$ can be directly derived using convex conjugation and proximal operator properties in terms of functions of the same form. Thus, in terms of theory, our can be thought of as a mild extension of the known proximal operator of the exponential function towards a regularized composition with a linear function. In terms of practice, we go "all the way through" from devising a naive formula that is correct but numerically problematic, to a reformulation in terms of known and reliable building blocks that do not suffer from numerical issues, and then towards a Python implementation.

The proximal operator of $x \to \ln(1 + \exp(x))$ is studied in Briceño-Arias et al. (2019), and is also derived in terms of similar functions. Similarly to this work, Briceño-Arias et al. (2019) attempt to provide a remedy to the numerical problem stemming from the exponentiation by devising a heuristic. However, in our work we tackle the numerical issue using established numerical tools rather than trying to re-invent the wheel.

A proximal operator for the so-called piece-wise exponential penalties $x \to 1 - \exp(-|x|/\rho)$ were studied in Liu et al. (2023; 2024). These works, too, obtained formulae that rely on the Lambert W function. Beyond the apparent similarity because of the use of the exponential function, the piecewise-exponential penalty is fundamentally different from the functions we study in this paper in equation 1. One is non-convex and approximates the $\ell_0$ "norm", whereas the other is convex.

Moreover, several works on incremental proximal point suggest using a simple bisection search for functions of the form $L(\langle \boldsymbol{\theta}, \boldsymbol{w} \rangle)$, where $L$ is a convex function, by reducing the proximal update to a solution of a scalar equation of one variable (Asi & Duchi, 2019; Toulis & Airoldi, 2014; Kulis & Bartlett, 2010). In this work we both deal with *regularized* losses with an additional squared Euclidean norm regularization, and propose a reduction to well-known special functions whose values can be computed much more efficiently and reliably with specialized algorithms devised by the numerical analysis community.

## 3   Preliminaries

Here, we recall some mathematical background, including the algorithmic framework from Shtoff (2024a). Then, we use the results to devise our proximal operator formula.

In this section we use concepts from convex analysis that are typically presented using the formalism of extended real-valued functions. Since in this paper we apply the preliminaries to convex functions defined on

the entire space, we specialize all preliminaries to regular convex functions to make this paper accessible to a wider audience without degrading its correctness.

## 3.1 Lambert's W function

Euler (Euler, 1783), based on the work of Lambert (Lambert, 1758), studied the solution set for $y$ of the equation

$$y \exp(y) = z$$

over the complex numbers. The solution set is described by a family of so-called Lambert W functions $W_k(z)$ for $k \in \{0, 1, 2, ...\}$. In this paper we focus on the solution set over the reals for $z > 0$, which corresponds to $W_0(z)$. See Corless et al. (1996) for a thorough introduction.

For simplicity, we denote $W_0(z)$ by $W(z)$, and refer to it as "the" Lambert W function. We note that Lambert W function is implemented in SciPy (Virtanen et al., 2020) as `scipy.special.lambertw`, in its full generality, for complex numbers and for any $k$.

Since the function $y \exp(y)$ grows faster than exponential as $y$ gets larger, its inverse, $W(z)$ can be thought of as a kind of a "logarithm". Indeed, it's occasionally called the Product-Log function, and it grows *slower* than a logarithm (Corless et al., 1996, eq 38):

$$W(z) = \ln(z) - \ln(\ln(z)) + o(1).$$

## 3.2 Moreau envelope

To describe the framework, we need to recall another concept related to the proximal operator - the Moreau Envelope (Moreau, 1962; 1965). For a function $r$, its Moreau envelope is:

$$M_r(\boldsymbol{w}) = \inf_{\boldsymbol{u}} \left\{ r(\boldsymbol{u}) + \frac{1}{2}\|\boldsymbol{u} - \boldsymbol{w}\|_2^2 \right\}.$$

Beyond the fact that the proximal operator is the minimizer of the same objective function, the Moreau envelope has another tight relation to the proximal operator, as shown in the following Lemma.

**Lemma 1** (Theorem 6.55 in Beck (2017)). *Let $r : \mathbb{R}^d \to \mathbb{R}$ be a convex function, and let $M_r$ be its Moreau envelope. Then $M_r$ is continuously differentiable, and its gradient is given by:*

$$\nabla M_r(\boldsymbol{w}) = \boldsymbol{w} - \text{prox}_r(\boldsymbol{w}) \tag{2}$$

## 3.3 Convex conjugate

The convex conjugate of the function $\psi$ is the function $\psi^*$ defined by

$$\psi^*(\boldsymbol{y}) = \sup_{\boldsymbol{u}} \left\{ \langle \boldsymbol{u}, \boldsymbol{w} \rangle - \psi(\boldsymbol{w}) \right\}.$$

The convex conjugate is a central object in optimization, and has many applications and properties. Moreover, tables of conjugate pairs for many useful functions have been devised in the literature. See Beck (2017) for in depth introduction and a comprehensive table of such functions, and the associated calculus properties. In particular, for the exponential function $\psi(t) = \exp(t)$ we have

$$\psi^*(s) = s \ln(s) - s,$$

defined for $s \geq 0$ with the convention that $0 \ln(0) \equiv 0$.

## 3.4 Algorithmic framework for regularized convex-on-linear proximal operator

The work of Shtoff (2024a) develops a generic framework for computing the proximal operator of functions of the the so-called *regularized convex-on-linear* form:

$$g(\boldsymbol{w}) = \psi(\langle \boldsymbol{\theta}, \boldsymbol{w} \rangle + b) + r(\boldsymbol{w}),$$

where $\psi$ is a univariate convex function, which typically has the role of a loss, and $r$ is a convex function that typically has the role of a regularizer. All the results below in this section were devised in the above work.

The resulting algortihmic framework for computing the proximal operator

$$\text{prox}_{\eta g}(\boldsymbol{w}) = \arg\min_{\boldsymbol{u}} \left\{ g(\boldsymbol{u}) + \frac{1}{2\eta} \|\boldsymbol{u} - \boldsymbol{w}\|_2^2 \right\}$$

comprises the following steps:

---

1. form the univariate function:

$$q(s) = M_{\eta r}(\boldsymbol{w} - \eta s\boldsymbol{\theta}) + (\langle \boldsymbol{\theta}, \boldsymbol{w} \rangle + b)s - \frac{\eta \|\boldsymbol{\theta}\|_2^2}{2} s^2 - \psi^*(s) \tag{3}$$

2. compute the unique maximizer $s^* = \arg\max q(s)$;

3. output $\text{prox}_{\eta g}(\boldsymbol{w}) = \text{prox}_{\eta r}(\boldsymbol{w} - \eta s^* \boldsymbol{\theta})$.

---

The function $q(s)$ is always concave. Oftentimes in practice it is also differentiable, and is maximized at the unique solution of $q'(s) = 0$. When $\psi^*$ is differentiable, so is $q$, and its derivative is given by:

$$q'(s) = \langle \boldsymbol{\theta}, \text{prox}_{\eta r}(\boldsymbol{w} - \eta s\boldsymbol{\theta}) \rangle - \psi^{*\prime}(s) + b. \tag{4}$$

For the derivative, the only computational tools that we require is the proximal operator of $r$, and the convex conjugate of $\psi$.

# 4 Deriving the closed-form formula

In this section we use the algorithmic framework for the proximal operator of regularized convex-on-linear functions to devise a closed-form formula for the function family in equation 1.

## 4.1 Using the algorithmic framework

The functions having the form in equation 1 are an instance of the family used by the algorithmic framework in section 3.4. Indeed, letting $\psi = \exp$, we can decompose the function $f$ as:

$$f(\boldsymbol{w}; \boldsymbol{\theta}, \boldsymbol{\phi}, b, \alpha) = \underbrace{\exp(\langle \boldsymbol{\theta}, \boldsymbol{w} \rangle + b)}_{\psi(\langle \boldsymbol{\theta}, \boldsymbol{w} \rangle + b)} + \underbrace{\langle \boldsymbol{\phi}, \boldsymbol{w} \rangle + \frac{\alpha}{2} \|\boldsymbol{w}\|_2^2}_{r(\boldsymbol{w})}. \tag{5}$$

Since we already know the convex conjugate $\psi^*(s) = s\ln(s) - s$, our missing ingredient is the proximal operator $\text{prox}_{\eta r}$. It is quite easy to derive, but it is just given in a variety of textbooks on optimization, such as Beck (2017, sec 6.2.3):

$$\text{prox}_{\eta r}(\boldsymbol{w}) = \frac{\boldsymbol{w} - \eta \boldsymbol{\phi}}{1 + \eta \alpha} \tag{6}$$

We are now ready to show how to maximize the function $q$ given in equation 3, that corresponds to the decomposition in equation 5.

**Lemma 2.** *The function $q(s)$ defined in equation 3 corresponding to the regularized convex-on-linear decomposition in equation 5 has a unique maximum $s^* > 0$ given by*

$$s^* = \frac{1}{\gamma} W(\exp(\delta + \ln(\gamma))),$$

$$\textit{where: } \gamma = \frac{\eta \|\boldsymbol{\theta}\|_2^2}{1 + \eta \alpha}, \tag{7}$$

$$\delta = \frac{\langle \boldsymbol{\theta}, \boldsymbol{w} - \eta \boldsymbol{\phi} \rangle}{1 + \eta \alpha} + b.$$

*Proof.* $\psi^*$ is differentiable, and its derivative is given by $\psi^{*\prime}(s) = \ln(s)$. Since $\psi^*$ is differentiable, so is $q(s)$, and by equation 4 its derivative is given by

$$
\begin{aligned}
q'(s) &= \langle \boldsymbol{\theta}, \text{prox}_{\eta r}(\boldsymbol{w} - \eta s \boldsymbol{\theta}) \rangle - \psi^{*\prime}(s) + b \\
&= \left\langle \boldsymbol{\theta}, \frac{\boldsymbol{w} - \eta s \boldsymbol{\theta} - \eta \boldsymbol{\phi}}{1 + \eta \alpha} \right\rangle - \ln(s) + b \\
&= -\frac{\eta \|\boldsymbol{\theta}\|_2^2}{1 + \eta \alpha} s - \ln(s) + \left\langle \boldsymbol{\theta}, \frac{\boldsymbol{w} - \eta \boldsymbol{\phi}}{1 + \eta \alpha} \right\rangle + b \\
&= -\gamma s - \ln(s) + \delta.
\end{aligned}
$$

We can see that $q'(s)$ is strictly decreasing on $s > 0$, and thus $q$ is strictly concave. Thus, if the equation $q'(s) = 0$ has a unique solution in $s > 0$, this must be the unique optimum. Adding $\ln(\gamma)$ to both sides of the equation $q'(s) = 0$ and simplifying, we obtain

$$
\gamma s + \ln(\gamma s) = \delta + \ln(\gamma).
$$

Exponentiating both sides, we get

$$
(\gamma s) \exp(\gamma s) = \exp(\delta + \ln(\gamma)).
$$

By definition of the Lambert W function, the above is equivalent to

$$
\gamma s = W(\exp(\delta + \ln(\gamma))),
$$

and therefore the unique solution is

$$
s^* = \frac{1}{\gamma} W(\exp(\delta + \ln(\gamma))),
$$

as required. $\qquad\square$

We now have a full algorithm for computing the proximal operator $\text{prox}_{\eta f}$ of functions $f$ having the form in equation 1:

1. Compute $s^*$ according to Equation equation 7,

2. Output:

$$
\text{prox}_{\eta f}(\boldsymbol{w}) = \frac{\boldsymbol{w} - \eta s^* \boldsymbol{\theta} - \eta \boldsymbol{\phi}}{1 + \eta \alpha}.
$$

Although it may appear that we are done, careful inspection of equation 7 shows that computing $s^*$ by definition requires computing $\exp(\delta + \ln(\gamma))$, which may lead to an overflow with floating point arithmetic. However, intuitively we understand that $W(x)$ acts as a kind of a logarithm, since it is the inverse function of $y \exp(y)$, and hence it should "cancel out" the effect of the exponentiation. The next section rigorously deals with this issue and provides a formula for computing $s^*$ that avoids exponentiation altogether.

## 4.2 Avoiding exponentiation with the Wright-Omega function

The Wright-Omega function $\omega(z)$ (Corless & Jeffrey, 2002) for a *real*[3] argument $z$ is defined by

$$
\omega(z) = W(\exp(z)).
$$

By definition of the Lambert W function, one can also see that $\omega(z)$ is the solution set of the equation

$$
y + \ln(y) = z,
$$

---

[3]Corless & Jeffrey (2002) present the definition for arbitrary complex numbers, but in this paper we specialize their definition to real numbers only

for $y$. Based on this observation, Lawrence et al. (2012) have devised a fast and direct numerical method for computing $\omega(z)$ *without* relying on the Lambert W function, and it has been implemented in SciPy as in `scipy.special.wrightomega`. Therefore, we can reformulate equation 7 for computing the solution of $q'(s) = 0$ as

$$s^* = \frac{1}{\gamma}\omega(\delta + \ln(\gamma)),$$

$$\text{where: } \gamma = \frac{\eta\|\boldsymbol{\theta}\|_2^2}{1 + \eta\alpha}, \tag{8}$$

$$\delta = \frac{\langle\boldsymbol{\theta}, \boldsymbol{w} - \eta\boldsymbol{\phi}\rangle}{1 + \eta\alpha} + b.$$

### 4.3 Open source implementation

We have created an open-source implementation of our proximal operator algorithm, both for NumPy arrays and PyTorch tensors. Both implementations support additional mini-batch dimensions prepended to the argument $\boldsymbol{w}$, and the parameters of the function $f$. Since PyTorch does not include an implementation of the Wright-Omega function, we have also ported the SciPy implementation to PyTorch in our code repository as well. The code can be found at `https://anonymous.4open.science/r/exponential-proximal-point-B8DD`, and includes tests to verify the correctness by comparing the resulting computation to the solutions obtained by CVXPY (Diamond & Boyd, 2016; Agrawal et al., 2018), and verifying that the gradient norms are close to zero. For completeness, we also provide our PyTorch Wright Omega implementation, and proximal operator implementation in Appendix **??**. Below is a description of our tests, and a summary of the results. Note, that the intention of our experiments is verifying that our proximal operator is correct and precise.

The input of the proximal operator are the parameters $\boldsymbol{\theta}, b, \boldsymbol{\phi}, \alpha$ of the function $f$ in equation 1, the parameter $\eta$ of the operator, and the point $\boldsymbol{w}$ at which we evaluate $\text{prox}_{\eta f}(\boldsymbol{w})$. Thus, we generate a set of $m$ test inputs $\{\boldsymbol{\theta}_i, b_i\boldsymbol{\phi}_i, \alpha_i, \eta_i, \boldsymbol{w}_i\}_{i=1}^m$ in the following manner:

$$\boldsymbol{\theta}_i, \boldsymbol{\phi}_i, \boldsymbol{w}_i \sim \text{Cauchy}(0, 1)^n,$$

$$b_i \sim \text{Cauchy}(0, 1),$$

$$\alpha_i, \eta_i \sim \text{LogNormal}(-3, 2),$$

where $n$ is the feature dimension. Our intention was to simulate features comprising of both small and large components, to challenge our implementation, and therefore we use the standard Cauchy distribution. Since $\alpha_i, \eta_i$ simulate step-size and regularization parameters, we simulated them from a log-normal distirbution that hopefully resembles realistic scenarios. Table 1 summarizes the statistics obtained from $m = 5,000$ samples of dimension $n = 50$:

Table 1: Percentiles of scalar and vector components in the simulated data

| Percentile Vector | P-0.1% | P-1% | P-25% | P-50% | P-75% | P-99% | P-99.9% |
|---|---|---|---|---|---|---|---|
| $\boldsymbol{w}$ | -294 | -31.5 | -0.992 | 0.00366 | 1 | 32.2 | 347 |
| $\eta$ | 0.000116 | 0.000468 | 0.0127 | 0.047 | 0.181 | 4.4 | 17.9 |
| $\boldsymbol{\theta}$ | -330 | -32.5 | -1.01 | -0.00377 | 0.999 | 32.3 | 334 |
| $\boldsymbol{\phi}$ | -341 | -32 | -1 | -0.00267 | 0.994 | 32.4 | 329 |
| $b$ | -138 | -26.1 | -0.986 | -0.0112 | 1.04 | 30.5 | 372 |
| $\alpha$ | 0.000126 | 0.000449 | 0.0124 | 0.0487 | 0.191 | 4.01 | 13.9 |

Since computing our $\text{prox}_{\eta f}(\boldsymbol{w})$ to minimizing a continuously differentiable convex function, an accurate solution must have a zero gradient of that function. Table 2 summarizes the statistics of gradient norms for various feature dimensions $n$ using $m = 10,000$ samples. We can see that even though the gradient norm grows with the dimension, which is expected, it is still almost zero for all practical purposes in almost all cases.

We note, however, that for very high-dimensional vectors, the worst-case norm is close to one, but looking the other statistics, it's not a common, and it may happen - our generated data is skewed and challenging on purpose.

Table 2: Gradient norm statistics of the proximal minimization objective for various feature dimensions $n$ using $m = 10,000$ samples.

| Dimension | Avg. | P-95% | Max. |
|----------:|------|-------|------|
| 10 | 1.23e-10 | 1.87e-11 | 2.18e-07 |
| 20 | 7.11e-09 | 4.00e-11 | 1.78e-05 |
| 50 | 1.67e-08 | 1.63e-10 | 5.40e-05 |
| 100 | 1.08e-08 | 5.04e-10 | 3.65e-05 |
| 500 | 1.37e-07 | 4.13e-09 | 3.42e-04 |
| 1000 | 3.69e-07 | 7.88e-09 | 8.65e-04 |
| 10000 | 1.85e-06 | 1.55e-07 | 2.44e-03 |
| 100000 | 2.93e-04 | 2.67e-06 | 9.14e-01 |

To appreciate how we compare to the default interior point solver of CVXPY, we plotted the gradiet norms of $m = 5,000$ samples of dimension $n = 50$ in Figure 1. We see that over-all, our solution is orders of magnitude more precise in a large portion of the cases. For some inputs, the CVXPY gradient norms show that the solution is pretty far from having a zero gradient.

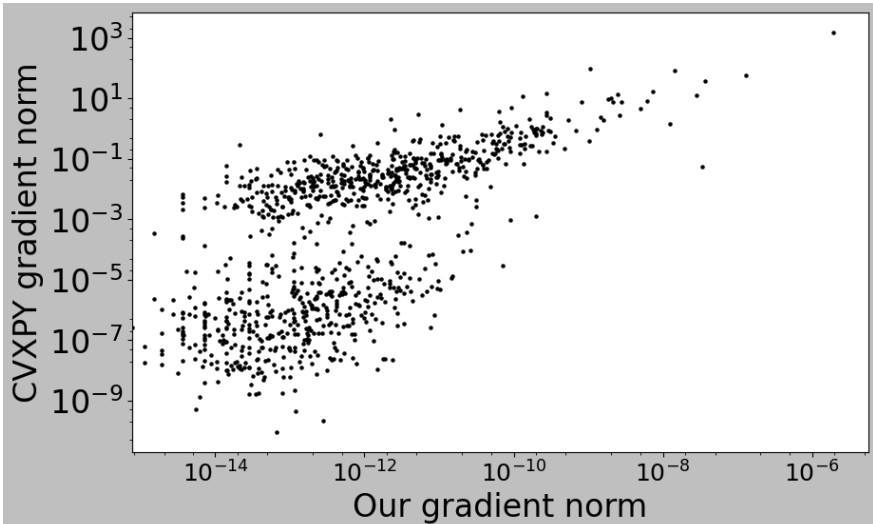

Figure 1: Comparison of the gradient norm of the proximal minimization objective of CVXPY and our method. Each point is one generated data-point. The X-Axis is the gradient norm obtined from our solution, whereas the Y-Axis is the gradient norm obtained from the CVXPY solution.

## 5 Summary

In this work we have devised a closed-form expression in terms of the Wright-Omega special function for the proximal operator for a family of functions that appears as regularized losses mainly in Poisson regression, but also in other possible applications. Our work allows researchers working incremental proximal point algorithms to perform numerical experiments with yet another machine-learning application of Poisson regression, and practitioners to use our work whenever their training procedure can potentially cause numerical overflows. We hope that in addition to other well-known "special" functions, such as the Gamma function, the Lambert W and the Wright-Omega function make it into additional machine-learning frameworks as first-class objects.

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

# A   Codes

## A.1   Wright Omega PyTorch implementation

```python
import torch

def wrightomega(x):
    """ Computes the Wright Omega function, based on the algorithm from SciPy:
    https://github.com/scipy/scipy/blob/maintenance/1.14.x/scipy/special/wright.cc#L369

    Example with special values:
    >>>specials = torch.tensor([float('nan'), float('inf'), float('-inf')])
    >>>wrightomega(specials)
    tensor([nan, inf, 0.])

    Example showing accuracy vs SciPy implementation
    >>>import numpy as np
    >>>import scipy
    >>>xs = np.r_[-np.flip(np.geomspace(1e-50, 1e30, 10000)), np.geomspace(1e-50, 1e30, 10000)]
    >>>scipy_results = scipy.special.wrightomega(xs)
    >>>torch_results = wrightomega(torch.tensor(xs))
    >>>torch.linalg.vector_norm(torch_results - scipy_results)
    tensor(1.7589e-13, dtype=torch.float64)

    """
    w = torch.zeros_like(x)
    w[x.isnan()] = float('nan')
    w[x.isinf() & (x > 0)] = float('inf')
    w[x.isinf() & (x < 0)] = 0
    finite_mask = torch.isfinite(x)

    tiny_mask = finite_mask & (x < -50)
    small_mask = finite_mask & (x >= -50) & (x < -2)
    w[small_mask | tiny_mask] = torch.exp(x[small_mask | tiny_mask])

    med_mask = finite_mask & (x >= -2) & (x < 1)
    w[med_mask] = torch.exp(2.0 * (x[med_mask] - 1.0) / 3.0);

    large_mask = finite_mask & (x >= 1) & (x < 1e20)
    lg = x[large_mask].log()
    w[large_mask] = x[large_mask] - lg + lg/x[large_mask];

    huge_mask = finite_mask & (x >= 1e20)
    w[huge_mask] = x[huge_mask]

    iterative_mask = small_mask | med_mask | large_mask

    # Iteration one of Fritsch, Shafer, and Crowley (FSC) iteration
    r = x[iterative_mask] - w[iterative_mask] - torch.log(w[iterative_mask]);
    wp1 = w[iterative_mask] + 1.0;
    e = (r / wp1) * (2.0 * wp1 * (wp1 + 2.0 / 3.0 * r) - r) / (2.0 * wp1 * (wp1 + 2.0/3.0*r) - 2.0 * r);
    w[iterative_mask] = w[iterative_mask] * (1.0 + e);

    finfo = torch.finfo(w.dtype)
    wp1 = torch.zeros_like(x).masked_scatter(iterative_mask, wp1)
    r = torch.zeros_like(x).masked_scatter_(iterative_mask, r)
    next_iter_mask = torch.abs((2.0**w*w-8.0**w-1.0)*torch.pow(torch.abs(r),4.0)) >= finfo.tiny*72.0*torch.pow(torch.abs(wp1), 6.0)
    iterative_mask = iterative_mask & next_iter_mask

    # FSC iteration two
    r = x[iterative_mask] - w[iterative_mask] - torch.log(w[iterative_mask]);
    wp1 = w[iterative_mask] + 1.0;
    e = (r / wp1) * (2.0 * wp1 * (wp1 + 2.0 / 3.0 * r) - r) / (2.0 * wp1 * (wp1 + 2.0/3.0*r) - 2.0 * r);
    w[iterative_mask] = w[iterative_mask] * (1.0 + e);

    return w
```

## A.2   PyTorch proximal operator implementation

```python
import torch
from numpy.typing import ArrayLike
from typing import Union
import numbers
from .wrightomega import wrightomega
```

```python
def is_scalar(x):
    if isinstance(x, numbers.Number):
        return True
    elif torch.asarray(x).ndim == 0:
        return True

    return False

def prox_op(w: ArrayLike,
            eta: Union[ArrayLike, float],
            theta: ArrayLike,
            phi: ArrayLike,
            b: Union[ArrayLike, float],
            alpha: Union[ArrayLike, float]):
    w = torch.as_tensor(w)
    eta = torch.as_tensor(eta)
    theta = torch.as_tensor(theta)
    phi = torch.as_tensor(phi)
    b = torch.as_tensor(b)
    alpha = torch.as_tensor(alpha)

    # broadcast input arguments expected to have one-less dimension than w, theta, and phi if they are not scalars
    if eta.ndim > 0:
        eta = torch.asarray(eta)[..., torch.newaxis]
    if b.ndim > 0:
        b = torch.asarray(b)[..., torch.newaxis]
    if alpha.ndim > 0:
        alpha = torch.asarray(alpha)[..., torch.newaxis]

    # compute formula parts
    common_denom = (1 + eta * alpha)
    gamma = eta * torch.sum(torch.square(theta), dim=-1, keepdim=True) / common_denom
    delta = torch.sum(theta * (w - eta * phi), dim=-1, keepdim=True) / common_denom + b

    # solve q'(s) = 0
    s = wrightomega(delta + torch.log(gamma)) / gamma

    # compute the result
    return (w - eta * s * theta - eta * phi) / common_denom
```

## A.3  NumPy / SciPy proximal operator implementation

```python
import numpy as np
from numpy.typing import ArrayLike
from typing import Union
from scipy.special import wrightomega

def prox_op(w: ArrayLike,
            eta: Union[ArrayLike, float],
            theta: ArrayLike,
            phi: ArrayLike,
            b: Union[ArrayLike, float],
            alpha: Union[ArrayLike, float]):
    w = np.asarray(w)
    theta = np.asarray(theta)
    phi = np.asarray(phi)

    # broadcast input arguments expected to have one-less dimension than w, theta, and phi if they are not scalars
    if not np.isscalar(eta):
        eta = np.asarray(eta)[..., np.newaxis]
    if not np.isscalar(b):
        b = np.asarray(b)[..., np.newaxis]
    if not np.isscalar(alpha):
        alpha = np.asarray(alpha)[..., np.newaxis]

    # compute formula parts
    common_denom = (1 + eta * alpha)
    gamma = eta * np.sum(np.square(theta), axis=-1, keepdims=True) / common_denom
    delta = np.sum(theta * (w - eta * phi), axis=-1, keepdims=True) / common_denom + b

    # solve q'(s) = 0
    s = wrightomega(delta + np.log(gamma)) / gamma

    # compute the result
    return (w - eta * s * theta - eta * phi) / common_denom
```

