# OpenReview forum: "Closed-form proximal operator of regularized exponential functions for incremental learning"
_TMLR — Rejected by TMLR_

### Review · Reviewer_xT2Q · 2024-12-04

**Summary Of Contributions:**

In this submission, the authors proposed a closed-form expression in terms of the Wright-Omega function for the proximal operator for a class of functions of regularized losses in Poisson regression with applications in large scale machine learning and data science problems. The authors also provide an open-source vectorized PyTorch implementation of the WrightOmega function and the proximal operator, ported from SciPy. This allows researchers to use the algorithm devised here to use the entire arsenal of tools provided by PyTorch, such as automatic differentiation and GPU computing.

**Audience:**

Yes

**Broader Impact Concerns:**

There are no concerns on the ethical implications of the work that would require adding a Broader Impact Statement.

**Claims And Evidence:**

Yes

**Requested Changes:**

All the request changes are from the weakness section:

1. The authors should provide results from numerical experiments in this submission.

2. The authors should add more explanations about the the Lambert's W function in section 3.1.

3. The authors should present the formal algorithm table in section 3.4 when they show the framework for regularized convex-on-linear proximal operator.

**Strengths And Weaknesses:**

This paper is organized with clean structure and clear language quality. The authors devised a closed-form formula for the proximal operator in terms of Lambert’s W function, whose implementation is available in many standard numerical computing and machine-learning packages, such as SciPy or TensorFlow. The authors provided all the theoretical proof and analysis for the mathematical formulas. The authors also created and published an open-source implementation of our proximal operator algorithm, both for NumPy arrays and PyTorch tensors.

However, this submission has the following weaknesses:

1. The theoretical contributions of this submission is not significant or impressive enough. The authors addressed the problems for a specific family of losses, which are a composition of exponential on linear functions. The application is Poisson regression. However, the model function can take different other forms from machine learning problems or data science problems. This submission only solve the problems with exponential forms with linear functions. This restrictions on the objective function forms make this submission only an incremental work. The authors should try to generalized the theoretical analysis to more general forms of the objective functions so that the impact or influence of this submission can be much more fundamental.

2. The authors should provide some empirical results from the numerical experiments in this submission. There are no numerical experiments section in this paper. Since the authors have published the implementation codes, it is natural to include some empirical results from these implementations in this work. This can significantly improve the quality of this submission if the empirical results are consistent with the theoretical analysis of this work.

3. The authors should present more detailed about the Lambert's W function in section 3.1. We need much more theories of the properties of this function in this submission. The current version of the contents of this Lambert's function is too short and brief in this work. The reader need more background and introduction of this function that plays essential roles in the theoretical analysis of this paper.

4. In section 3.4, the authors should present the algorithmic framework for regularized convex-on-linear proximal operator as the formal algorithm form. The authors should provide the detailed algorithm with the forms of Algorithm table in this submission.

---

> ### Author Response · Authors · 2024-12-08
> **Author response**
>
> # Regarding weaknesses
> ## W1
> The more general family of losses, and additional families, is addressed by the cited paper by Shtoff (2024), where the algorithmic framework comes from. That paper is the one with the  "more fundamental impact".  The algorithms there are less efficient, and are similar to a one-dimensional bisection search in nature. This paper, intentionally, is not about genericity, but specificity. We sacrifice genericity in the sake of efficiency by utilizing well-studied special functions that have extremely fast and efficient algorithms for evaluating them. To the best of our knowledge, TMLR's acceptance criteria are exactly about that - rigor, correctness, and interest to some part of the machine learning community.
>
> ## W2
> We note that our algorithmic contribution is not "yet another method to train models'', but "yet another computational tool for existing methods to train models''. Hence, numerical experiments showing that the tool "works", namely, computes the proximal operator close to machine precision according to some error metric, make a lot of sense. As you said, the code is there, and we can add them.
>
> However,  experiments reproducing the results shown by the cited papers by Asi, Duchi, Davis, and Drusvyatsky, demonstrating the robustness of proximal point methods to step size choice, are not the focus of this paper. As we said, this paper is not "yet another method to train models''.
>
> Overall, if the reviewer ment experiments demonstrating that the computational tool works, we strongly agree with the reviewer, and will gladly add them. If the reviewer meant experiments reproducing already shown results about the robustness of proximal point methods, we strongly disagree here.
>
> ## W3
> There is only *one* property of the Lambert W function needed for this paper, and this is the fact that it is a "kind of a logarithm''. We agree that adding it formally, rather with hand waving, would be better. For example, we could add the approximation
> $$
> W_0(x) \approx \ln(x) - \ln(\ln(x))
> $$
> that stems from the asymptotic expansion of $W_0(x)$, or the equality
> $$
> W_0(x) = \ln \left( \frac{x}{W_0(x)} \right),
> $$
> that holds globally for $x \geq -\frac{1}{e}$, to show that Lambert W has near logarithmic growth.
>
> ## W4
> If the reviewer means using an Algorithm LaTeX environment, then we shall do it. Indeed it would look better.

---

### Review · Reviewer_bEBX · 2024-12-14

**Summary Of Contributions:**

The authors present a closed-form proximal operator for an exponential function of a linear term with $\ell^2$ regularization, using Lambert’s $W$ function. They extend this to the Wright-Omega function, which avoids floating-point overflow. Finally, they provide an implementation for efficiently computing these proximal operators.

**Audience:**

Yes

**Claims And Evidence:**

Yes

**Requested Changes:**

- I think the authors should cite this [website](https://proximity-operator.net/scalarfunctions.html#:~:text=al.%2C%202011%5D-,Exponential,-exp), which provides a closed-form formula for the exponential function, and cite the reference provided there.
- As mentioned in Section 4.3, where you note a comparison to existing methods, including plots for this comparison would enhance the paper.
- The implementation code could also be included in the paper, along with a clear explanation of each line.

---

Typos:
- In the equation below Equation (2), it should likely be $w$ instead of $y$ on the left-hand side.
- Below page 4, $q$ is concave, not $q^\prime$. Additionally, it should be $\psi^*$ instead of $h^*$.
- In Section 2 (Related Work), it should be $x \to$ instead of $t\to$.

**Strengths And Weaknesses:**

Strengths:
- The paper is well-written and provides a clear, explicit derivation of the proximal operator for an exponential function.
- The use of the Wright-Omega function effectively avoids floating-point overflow.
- It provides an implementation.

---

Weaknesses:
- The contribution feels more like solving a textbook problem. The formula for the proximal operator of an exponential function is already publicly available [here](https://proximity-operator.net/scalarfunctions.html#:~:text=al.%2C%202011%5D-,Exponential,-exp). The paper cited on the website does not provide a derivation of this formula, so this work adds value by offering a derivation.
- In Section 2 (Related Work), you state, “One is non-convex, whereas the other is convex.” The function in equation (1) is convex, as is $1-\exp(-|x| / \sigma)$. Which functions are you referring to as non-convex? How do the derivations in the previous work differ, and why can’t those methods be directly applied here?

---

> ### Author Response · Authors · 2024-12-20
> **Comments**
>
> ### Weaknesses
> - It is what it is - we cannot hide it. This paper is a computational tool for existing frameworks for training models, rather than a new famework or a new analysis. We hope that the revised Section 4.3 makes a more convincing argument that the computational tool 'works'. We believe there's a value in taking a problem and going 'all the way through' to obtain a reliable and working solution.
> - The $1 - \exp(-|x|/\sigma)$ is non-convex. It's an approximation of the $\ell_0$ penalty for promoting sparsity. Its plot looks like this (the asterisks are the plot):
> ```
>       +---------------------------------------+
>     1 |***************         ***************|
>   0.8 |              ***     ***              |
>   0.4 |                **   **                |
>   0.2 |                 ** **                 |
>     0 |                  ***                  |
>       +---------------------------------------+
>      -6     -4    -2      0      2     4      6
> ```
> Beyond the use of $\exp(\cdot)$ and the use of Lambert W in its proximal operator, this function shares no similarities to the one studied in this paper.
>
> Beyond that, we'd like to point out that we believe this work conforms to the TMLR acceptance criteria: correct, and interesting to some subset of the ML community. The scale of its impact, weather big or small, is to the best of our knowledge among the criteria.

---

### Review · Reviewer_u2mH · 2024-12-15

**Summary Of Contributions:**

This paper presents a closed-form formula for the proximal operator of regularized exponential functions, expressed using the Lambert $W$ function and reformulated with the Wright-Omega function for improved numerical stability. Additionally, it offers an open-source implementation in PyTorch and SciPy, with support for GPU acceleration. These contributions address computational challenges in proximal-point methods and are particularly relevant to applications like Poisson regression.

**Audience:**

Yes

**Claims And Evidence:**

Yes

**Requested Changes:**

P2, paragraph beginning with "When f is the Taylor approximation", what are $\ell$ and $h$? Please specify.

**Strengths And Weaknesses:**

The paper derives a closed-form proximal operator for $f$ as in (1), building directly on the framework established by Shtoff (2024a). While it provides a specific implementation leveraging the Lambert $W$ and Wright-Omega functions, the contribution appears very limited and incremental as it primarily applies existing theoretical tools to a specific function family. This limited scope may not represent a substantial enough advance for a scientific publication, as it largely refines and implements prior methods rather than introducing any novel ideas.

---

> ### Author Response · Authors · 2024-12-20
> **Comment**
>
> Thanks for pointing out the typo. It has been fixed in the latest revision.
>
> Regarding the scope - it is what it is. It's about devising a (hopefully) reliable computational tool for an existing framework of algorithms, the proximal-point algorithms, rather than creating new frameworks and studying their convergence properties. I hope that the plots and tables in Section 4.3 in the revision make a more convincing argument about the reliability of the tool.
>
> It appears to conform to TMLR criteria: correct, and interesting to a subset of the ML community. It's not about the amount of impact the contribution makes.

---

### Comment · Reviewer_u2mH · 2024-12-15
**Contributions Too Limited for Publication**

The paper derives a closed-form proximal operator for $f$ as in (1), building directly on the framework established by Shtoff (2024a). While it provides a specific implementation leveraging the Lambert $W$ and Wright-Omega functions, the contribution appears very limited and incremental as it primarily applies existing theoretical tools to a specific function family. This limited scope may not represent a substantial enough advance for a scientific publication, as it largely refines and implements prior methods rather than introducing any novel ideas.

---

### Author Response · Authors · 2024-12-20
**Revised version**

I have uploaded a revised version, in accordance to the reviews. Here is a summary of the changes:
- Cited proximal operator of exponential function in the related work (referred from https://proximity-operator.net/scalarfunctions.html#:~:text=al.%2C%202011%5D-,Exponential,-exp). I have not cited the website, but the papers this website refers to that contain the results.
- No need for a "Lemma" for the proximal operator of the `regularizer'. It's a simple quadratic, and there's a known formula, and it's a few lines to derive. There is a limit to how much I want the paper to be 'self contained'.
- Numerical accuracy experiments (section 4.3)
- Add implementation code as an appendix to the paper.
- Use a property of the Lambert W function to show why it is a `kind of a logarithm'
- Typos pointed out by the reviewers

---

### Decision · Action_Editor_wpuP · 2025-01-31

**Recommendation:** Reject

**Comment:**

This work derives the "closed-form" solution to the proximal operator of the exponential function. The AE puts a quotation mark because, while the connection to the Lambert's W function and the Wright-Omega function is intriguing, its calculation is still an iterative procedure.

It might be of the authors' interest to check $\S3$ of the paper *Stochastic Douglas-Rachford Splitting for Regularized Empirical Risk Minimization: Convergence, Mini-batch, and Implementation* published in TMLR 2022. Using the authors' notation, if one is interested in calculating the proximal operator of $\psi(\theta^T w+b)$ at $u$, then the operator is always in the form $u+\alpha\theta$, where $\alpha$ is some unknown scalar. Plugging it back into the proximal objective leads to a scalar optimization problem, which can be solved numerically via bisection if $\psi$ is convex. It is not as elegant as the result of this work, but fundamentally the goal is to calculate a certain scalar value (in this paper is $s^*$ in (7)) via some iterative procedure.

One of the reviewers provided the link to a website showing that this result, which involves the Lambert's W function, is not completely unknown in the field. The proof of the claim is quite elementary, giving readers the impression of a "textbook exercise".

**Audience:**

No.

Reviewers pointed out that the result is quite elementary and not unknown in the field.

**Claims And Evidence:**

Yes.